# Prospective Assessment of Cerebral Microbleeds with Low-Field Magnetic Resonance Imaging (0.55 Tesla MRI)

**DOI:** 10.3390/jcm12031179

**Published:** 2023-02-02

**Authors:** Thilo Rusche, Hanns-Christian Breit, Michael Bach, Jakob Wasserthal, Julian Gehweiler, Sebastian Manneck, Johanna M. Lieb, Gian Marco De Marchis, Marios Psychogios, Peter B. Sporns

**Affiliations:** 1Department of Radiology, Clinic of Radiology & Nuclear Medicine, University Hospital of Basel, University of Basel, 4001 Basel, Switzerland; 2Imamed Radiologie Nordwest AG, 4051 Basel, Switzerland; 3Department of Radiology, Gesundheitszentrum Fricktal, 4310 Rheinfelden, Switzerland; 4Department of Neurology, University Hospital of Basel, University of Basel, 4001 Basel, Switzerland; 5Department of Diagnostic and Interventional Neuroradiology, University Medical Center Hamburg-Eppendorf, 20251 Hamburg, Germany; 6Department of Radiology and Neuroradiology, Stadtspital Zürich, 8063 Zürich, Switzerland

**Keywords:** low-field MRI, MRI, reading study, scanner comparison, cerebral microbleeds, 0.55 T

## Abstract

Purpose: Accurate detection of cerebral microbleeds (CMBs) on susceptibility-weighted (SWI) magnetic resonance imaging (MRI) is crucial for the characterization of many neurological diseases. Low-field MRI offers greater access at lower costs and lower infrastructural requirements, but also reduced susceptibility artifacts. We therefore evaluated the diagnostic performance for the detection of CMBs of a whole-body low-field MRI in a prospective cohort of suspected stroke patients compared to an established 1.5 T MRI. Methods: A prospective scanner comparison was performed including 27 patients, of whom 3 patients were excluded because the time interval was >1 h between acquisition of the 1.5 T and 0.55 T MRI. All SWI sequences were assessed for the presence, number, and localization of CMBs by two neuroradiologists and additionally underwent a Likert rating with respect to image impression, resolution, noise, contrast, and diagnostic quality. Results: A total of 24 patients with a mean age of 74 years were included (11 female). Both readers detected the same number and localization of microbleeds in all 24 datasets (sensitivity and specificity 100%; interreader reliability ϰ = 1), with CMBs only being observed in 12 patients. Likert ratings of the sequences at both field strengths regarding overall image quality and diagnostic quality did not reveal significant differences between the 0.55 T and 1.5 T sequences (*p* = 0.942; *p* = 0.672). For resolution and contrast, the 0.55 T sequences were even significantly superior (*p* < 0.0001; *p* < 0.0003), whereas the 1.5 T sequences were significantly superior (*p* < 0.0001) regarding noise. Conclusion: Low-field MRI at 0.55 T may have similar accuracy as 1.5 T scanners for the detection of microbleeds and thus may have great potential as a resource-efficient alternative in the near future.

## 1. Introduction

Cerebral microbleeds (CMBs) are small (2–10 mm diameter), round, or ovoid hypointense foci with associated blooming with enhanced visibility on MRI sequences sensitive to susceptibility artifacts [1,2]. They can be observed in patients with cognitive complaints and stroke but also in healthy individuals. Technically, local magnetic field inhomogeneities caused by paramagnetic iron in CMBs result in signal loss on MRI sequences, such as T2*-weighted gradient echo sequences or susceptibility-weighted imaging (SWI). SWI is derived from gradient echo sequences with additional post-processing to improve contrast resolution and is usually acquired in three dimensions to increase spatial resolution with flow compensation in all three planes to reduce artifacts. SWI has increased sensitivity and reliability for CMBs compared with T2*-weighted gradient echo but requires a longer acquisition time [3]. Further, technical aspects such as a low flip angle, long echo time, and long repetition time increase the sensitivity to susceptibility effects [2]. Moreover, the susceptibility effect and signal-to-noise ratio have been described to increase with higher magnetic field strength [4].

On the other hand, MRI accessibility is low and extremely inhomogeneous around the world, because MRI installations require expensive infrastructure (e.g., site preparation to host the large magnets, magnetic/radiofrequency shielding, and emergency helium exhaust conduit), high maintenance costs (i.e., for helium refill), have high operational costs for specialized radiographic technicians, and require huge amounts of electricity and water leaving a large ecological footprint [5,6]. Thus, the distribution of MRI scanners is concentrated mainly within high-income countries, and ~70% of the world’s population has little to no access to MRI (OECD (2022), magnetic resonance imaging (MRI) units (indicator); DOI: 10.1787/1a72e7d1-en). Moreover, even in high-income countries, clinical MRI scanners are mostly located in highly specialized radiology departments, large and centralized imaging centers, and housed on the ground floors of hospitals and clinics, excluding easy access to neurology clinics, trauma centers, surgical suites, neonatal/pediatric centers, and community clinics [5]. With this in mind, low-field MRI is increasingly coming into focus, offering MR imaging at a much lower cost, offering a considerable energy-saving potential but also reducing possible complications with metallic implants. Therefore, the aim of our study was to evaluate the performance of a 0.55 T low-field MRI in a prospective cohort of suspected stroke patients and to directly compare the diagnostic value for the detection of CMBs to a standard 1.5 T MRI.

## 2. Materials and Methods

This prospective study was reviewed and approved by the cantonal (Basel-Stadt, Switzerland) ethics committee (BASEC2021-00166). All included patients signed an informed consent form.

### 2.1. Data Acquisition

Data acquisition was performed from 1 May 2021 to 30 June 2021 at the Division of Neuroradiology, Clinic of Radiology and Nuclear Medicine, University Hospital Basel, Switzerland. All patients who presented to the emergency room with suspected ischemic stroke or transient ischemic attack (TIA) and who underwent MRI using a 1.5 T scanner (Siemens MAGNETOM Avanto FIT 1.5 T; Siemens Healthineers; Erlangen, Germany) as part of the diagnostic stroke workup were included. Immediately afterwards, patients were examined using a 0.55 T scanner (Siemens MAGNETOM FreeMax 0.55 T; Siemens Healthineers; Erlangen, Germany). The 1.5 T scanning protocol was in accordance with the hospital’s internal standard protocol for emergency stroked diagnostics including SWI sequences (Table 1). The 0.55 T SWI protocol was adapted to the 1.5 T SWI protocol as far as technically possible (same slice thickness (ST) and slice spacing (SP); comparable in-plane resolution) to ensure the most objective scanner comparison. After subsequent verification with respect to data completeness (scan protocols with complete image acquisition) and image quality (artifacts, image contrast), the datasets were transferred to the Picture Archiving and Communication System (PACS; General Electric (GE); Waukesha, WI, USA) for further analysis.

### 2.2. Data Analysis

Data analysis was performed in a two-step procedure: First, the 0.55 T and 1.5 T SWI datasets were evaluated using Likert rating. Second, a reading study was performed regarding identification, localization, and number of CMBs.

### 2.3. Likert Rating

Likert rating was performed by a neuroradiologist and a neuroradiologist in training with an experience of 9 and 5 years. Each acquired 0.55 T or 1.5 T SWI dataset was rated with respect to the following criteria with a numerical value between 1 and 10 (1 worst, 10 best):

(a)Overall image quality;(b)Resolution;(c)Noise;(d)Contrast;(e)Diagnostic quality.

Sample SWI sequences from a 3 T scanner (Siemens MAGNETOM Skyra 3 T; Siemens Healthineers; Erlangen, Germany) were set as the gold standard (numerical value = 10). Dataset assessment was PACS-based using a standardized bookmark. Both readers were blinded to the results of the other reader.

### 2.4. Reading Study

Reading of 0.55 T and 1.5 T datasets was performed PACS-based and blinded (no clinical information, no image information) by two neuroradiologists with 8 and 13 years of professional experience. PACS-based post-processing as part of image analysis was allowed. For each dataset, number of SWI lesions (0, 1, 2–10, >10) and SWI lesion localization were analyzed.

The final neuroradiological report was defined as the underlying gold standard for the accuracy of the reading study.

## 3. Statistical Analysis

For statistical evaluation of Likert rating, a mean of the ratings of Readers 1 and 2 was first calculated for each 0.55 T and 1.5 T patient dataset and evaluation points (a)–(e). Subsequently, a Wilcoxon signed-rank test was used to evaluate significant or non-significant differences in Likert rating between the 0.55 T and 1.5 T sequences. Then interreader comparisons were performed to determine intraclass correlation coefficients (ICC). Calculation of sensitivity and specificity of Readers 1 and 2 in the reading study was performed in relation to the gold standard.

## 4. Results

A total of 27 patients with complete and artifact-free datasets were prospectively included in this study, of whom 3 patients were excluded because the time interval between the 1.5 T scan and 0.55 T scan was >1 h. The mean age of the remaining 24 patients was 74 years (standard deviation 14 years), and 11 patients were female (46%)**.** The mean time interval between the 1.5 T scans and 0.55 T scans was 36 ± 14 min. Baseline characteristics of the patient cohort are presented in Table 2.

Both readers detected the same number and localization of microbleeds in all 24 0.55 T and 1.5 T datasets (sensitivity and specificity 100%; interreader reliability ϰ = 1). Ten 1.5 T datasets did not contain any microbleeds by analysis of the more experienced neuroradiologist and were therefore defined as the control group. No false-positive findings were observed by assessment of these images by Reader 2 (positive predictive value and negative predictive value 100%).

Likert ratings of the sequences with both field strengths regarding overall image quality (a) and diagnostic quality (e) did not reveal significant differences between the 0.55 T and 1.5 T sequences (*p* = 0.942 and *p* = 0.672, respectively; see Figure 1). Regarding the subjective evaluation of the spatial resolution (b) and contrast (d), the 0.55 T sequences were rated to be significantly superior: (b) *p* < 0.0001; (d) *p* < 0.0003. In contrast, the 1.5 T sequences were superior (*p* < 0.0001) regarding noise (c). Interreader comparisons showed moderate to high levels of agreement between Reader 1 and Reader 2 for the Likert ratings (ICC: (a) 0.91; (b) 0.93; (c) 0.60; (d) 0.87; (e) 0.88).

## 5. Discussion

Our study shows that detection of microbleeds in SWI MRI sequences at a field strength of 0.55 T is possible with the same specificity and sensitivity compared to conventional 1.5 T MRI (for sample sequences, see Figure 2). Moreover, Likert ratings for the subjective evaluation of spatial resolution and contrast resolution were not significantly different for both 0.55 T and 1.5 T sequences, whereas for SWI noise, 0.55 T sequences were even superior.

To the best of our knowledge, no comparable study has performed a 0.55 T versus 1.5 T scanner comparison regarding the detection of microbleeds before. In a previous study, we showed that low-field MRI is not inferior to scanners with higher field strength for the detection of small infarcts in DWI and FLAIR sequences; however, in this study, there were minor limitations in the detection of very small infarcts [7]. Another group evaluated the performance of a modern 0.55 T MRI in the diagnosis of intracranial aneurysms in comparison to the gold standard digital subtraction angiography (DSA) [8]. This study included a total of 19 aneurysms in 16 patients, which were identified in both 0.55 T magnetic resonance angiography and DSA. Moreover, measurements of the two readers showed no significant differences between 0.55 T TOF MRA and DSA in the overall aneurysm size (calculated as the mean from height/width/neck), as well as in the mean width and neck values. The mean height was significantly larger in 0.55 T TOF MRA in comparison to DSA, whereas intermodality (1.5 T and 3 T TOF MRA) and interrater agreement were excellent (ICC > 0.94). Thus, the authors concluded that TOF MRA acquired with a modern 0.55 T MRI is a reliable tool for the detection and initial assessment of intracranial aneurysms. Moreover, in another study, we showed that patients perceived 0.55 T new-generation low-field MRI to be more comfortable than conventional 1.5 T MRI, given its larger bore opening and reduced noise levels during image acquisition, and concluded that new concepts regarding bore design and noise level reduction of MR scanner systems may help to reduce patient anxiety and improve well-being when undergoing MR imaging [9]. For microbleeds, the diagnostic accuracy of SWI sequences of the 0.55 T Magnetom FreeMax in our study seemed to be even higher, offering great potential for the characterization of associated diseases such as diffuse axonal injury or cerebral amyloid angiopathies [8].

In principle, the use of low-field MRI in clinical routine has several advantages. First, accessibility is low and extremely inhomogeneous around the world because MRI installations require expensive infrastructure, so the distribution of MRI scanners is concentrated mainly within high-income countries, and ~70% of the world’s population has little to no access to MRI [5]. Moreover, even in high-income countries, clinical MRI scanners are mostly located in highly specialized radiology departments, large and centralized imaging centers, and in those are housed on the ground floors of hospitals [5]. In a recent analysis by our group, we could show that in terms of purchase price, the savings potential of a 0.55 T MRI compared to a 1.5 T MRI system is about 40–50% [10]. The 25% lower weight of the system additionally reduces the transportation costs incurred, and the smaller size of the unit allows for installation by a remotely controlled mobile robotic system without opening the exterior façade, if the operating site is at ground level. Together with the lack of need to install a quench pipe, this reduces the total cost of installation by up to 70%. The maintenance cost of a 0.55 T MRI is approximately 45% less than that of a 1.5 T unit with a comparable service contract. Further cost reductions result from the smaller room size and potentially lower energy consumption for examinations and cooling. In conclusion, the use of lower-field-strength MRI systems offers enormous economic and environmental potential for both hospitals and practice operators, as well as for the healthcare system as a whole. In this context, offering MR imaging at a lower cost and with fewer infrastructural requirements will be key to increasing access to MRI for many patients. In developed countries, the energy-saving potential [6], and the possibility to reduce complications and artifacts caused by metallic implants will be the major arguments for implementing low-field MR imaging. The increasing exploration of the potential and limitations of low-field MRI is crucial to guide this implementation at a larger scale without harming patients.

### Limitations

There are several limitations that need to be addressed. First, a 1.5 T device of routine clinical use is in fact not the “gold standard” for the detection of CMBs. Various studies have shown that higher field strengths (3 T and 7 T) have the highest sensitivity for the detection of CMBs [2,4,11,12,13,14,15,16,17]. For example, Conijn et al. [4] showed that the detection of CMBs is more reliable at 7 T compared to 1.5 T. Stehling et al. [18] had similar results when comparing 1.5 T versus 3 T. Greenberg et al. [2] also described better CMB detection at higher field strengths (3 T or higher), as CMBs are better visible in this case due to stronger susceptibility artifacts and thus stronger blooming artifacts. These results were most recently supported by data from Bian et al. [14], who also confirmed a higher sensitivity in the diagnosis of radiotherapy-induced CMBs at 7 T compared to 3 T sequences. For this reason, the detection of CBMs with lower field strengths (0.55 T and 1.5 T) is already a priori suboptimal, and thus the definition of a 1.5 T scanner as a comparison scanner and “gold standard” is afflicted with deficiencies. Thus, future studies should compare 0.55 T versus 3 T or 7 T. Second, both scanners differ with respect to their gradient and coil system as well as the field strength. Third, the study cohort (only 12 patients with CMBs), nonetheless prospective, is still relatively small. Larger-scale studies to further define indications for the detection of microbleeds at 0.55 T are needed and should assess whether scanner choice has an impact on patient outcomes. Fourth, results may have been different if the study cohort would have been selected from a population where a high CMB burden is already likely due to expected or diagnosed underlying diseases (cerebral amyloid angiopathy (CAA) or Alzheimer’s disease). Thus, more CMBs would be detectable and comparable in the collective as a whole.

## 6. Conclusions

Low-field MRI at 0.55 Tesla may have the same accuracy as 1.5 T MRI for the detection of microbleeds and thus may have great potential as a low-cost alternative in the near future.

## Figures and Tables

**Figure 1 jcm-12-01179-f001:**
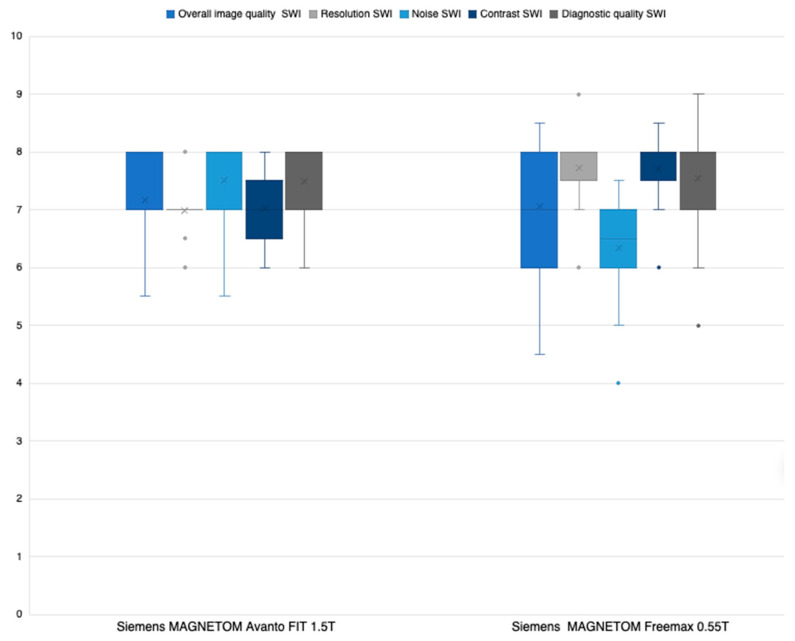
Average Likert scoring of Readers 1 and 2 SWI sequences.

**Figure 2 jcm-12-01179-f002:**
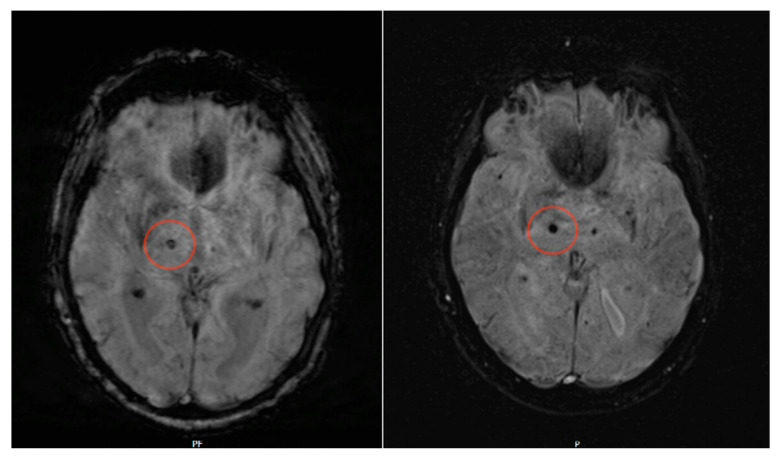
SWI lesion of the right thalamus. In both images (left, axial 1.5 T SWI sequence; right, axial 0.55 T SWI sequence), this lesion is clearly detectable, although it is even better delineated in the 0.55 T dataset.

**Table 1 jcm-12-01179-t001:** Scan protocols Siemens MAGNETOM Avanto FIT 1.5 T and Siemens MAGNETOM FreeMax 0.55 T.

	Siemens MAGNETOM FreeMax 0.55 T	Siemens MAGNETOM Avanto Fit 1.5 T
FLAIR tra		
Field strength in T	0.55	1.5
Field of view (FOV) in mm^2^	209 × 230	187 × 230
Slice thickness (ST) in mm	3	3
Slice spacing (SS)	3.6	3.6
Number of slices	40	40
Pixel spacing (PS) in mm^2^	1.28 × 1.03	0.9 × 0.9
Repetition time (TR) in msec	7780	8510
Echo time (TE) in msec	96	112
Inversion delay (TI) in msec	2368.8	2460
Turbo factor	15	19
Time of acquisition (TA) in min	05:28	03:26
BW ((BW))	150	130
3D SWI tra		
Field strength in T	0.55	1.5
Sequence type	Multi-shot 3D EPI	3D FLASH
Field of view (FOV) in mm^2^	201 × 230	194 × 230
Slice thickness (ST) in mm	3	3
Number of slices	40	48
Pixel spacing (PS) in mm^2^	0.94 × 0.8	1.12 × 0.9
Repetition time (TR) in msec	172	48
Echo time (TE) in msec	100	40
Parallel imaging	-	GRAPPA factor 2
Time of acquisition (TA) in min	02:23	02:17
BW ((BW))	276	80
Single-shot diffusion EPI tra		
Field strength in T	0.55	1.5
Field of view (FOV) in mm^2^	220 × 220	230 × 230
Slice thickness (ST) in mm	3	3
Slice spacing (SS)	3.6	3.6
Number of slices	40	40
Pixel spacing (PS) in mm^2^	1.67 × 1.67	1.44 × 1.44
b-values in s/mm^2^	0, 1000	0, 1000
Repetition time (TR) in msec	7400	6200
Echo time (TE) in msec	102	103
Parallel imaging	GRAPPA factor 2	GRAPPA factor 2
Time of acquisition (TA) in min	04:35	02:04
BW ((BW))	842	1490

**Table 2 jcm-12-01179-t002:** Detailed patient data.

Patient	Patient Age	CMB Yes/No	Number of CMB	(Main)-Localization of CMB	Time Gap between Scans in min
Patient 1	87	Yes	1	Left occipital	46
Patient 2	73	Yes	3	Right frontal/periventricular	25
Patient 3	88	Yes	4	Right temporal/parietal	37
Patient 4	29	No	0	-	33
Patient 5	82	No	0	-	93
Patient 6	70	No	0	-	44
Patient 7	87	Yes	2	Left occipital	32
Patient 8	74	No	0	-	25
Patient 9	60	No	0	-	21
Patient 10	44	No	0	-	49
Patient 11	84	Yes	1	Left occipital	33
Patient 12	58	No	0	-	40
Patient 13	80	No	0	-	35
Patient 14	65	Yes	1	Left Putamen	20
Patient 15	65	Yes	2	Left frontal	24
Patient 16	75	No	0	-	22
Patient 17	84	No	0	-	48
Patient 18	82	Yes	4	Left frontal	32
Patient 19	79	Yes	1	Left occipital	42
Patient 20	84	Yes	1	Left periventricular	32
Patient 21	86	Yes	>10	Bilateral Thalamus	25
Patient 22	83	No	0	-	31
Patient 23	89	No	0	-	38
Patient 24	69	Yes	3	Left periventricular	42
Patient 25	53	Excluded	-	-	916
Patient 26	59	Excluded	-	-	2936
Patient 27	46	Excluded	-	-	2812

## Data Availability

The data presented in this study are available on request from the corresponding author.

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
