# Peer review of "Prospective Assessment of Cerebral Microbleeds with Low-Field Magnetic Resonance Imaging (0.55 Tesla MRI)"

_jcm, 2023, doi:10.3390/jcm12031179_

Round 1

Reviewer 1 Report

The authors compared 1,5T MRI to 0,55T MRI in terms of detection and characterisation of microbleeds in a prospective cohort of 24 patients and found that low-field MRI showed the same accuracy as 1,5T MRI. The manuscript is well written and I have no substantial comments.  Please state in the abstract that microbleeds were actually detected in 12 out of 24 patients.

Author Response

Please see the attachment (Word-doc). 

Reviewer 2 Report

In this manuscript, Rusche et al evaluated  the detection off cortical microbleeds on 0.55T and 1.5T in 24 elderly patients. They found that all CMBs detected on 1.5T were also detected on 0.55T MRI. They conclude that  low fied mRI at 0.55T has the same accuracy as scanners with higher field strengths for the detection fo CMBs.

This is a relatively simple study with potential clinical importance but the conclusions drawn are very overstated.

Firstly , multiple prior studies have shown that CMB detection is better with greater field strength.  According to Greenberg et al “Increased magnetic field strength, 3 Tesla (T) or even higher, appears to improve CMB conspicuity.23-26 Susceptibility effect is higher at higher magnetic field strength and blooming effects are therefore predicted to be greater” Lancet Neurol. 2009 February ; 8(2): 165–174. doi:10.1016/S1474-4422(09)70013-4.

So I am not certain that comparing two relatively low field magnetic strengths which are both considered suboptimal for CMB detection allows the authors to make such assertive conclusions about the ability of 0.55t to detect CMB at similar accuracy to scanners with higher field strength. 

The only conclusion that can be deduced from this study is that 0.55T MAY  have similar accuracy to 1.5T MRI for detection of CMBs as opposed to a blanket statement of MRIs of increased higher field strength. It is also problematic coming to these strong statements given that only 11 patients had CMB in this study. An ideal study sample will also have involved populations expected to have high CMB burden such as those with suspected CAA or patients with alzheimer’s disease.  The conclusion needs to be revised to reflect these changes.

Moreover given that field strength is one of the strongest determinants of CMB detection, the authors need to discuss extensively  (1) Summarize results of prior studies that have demonstrated increased CMB detection with increased field strength (2) Discuss extensively why their results contradict those of others that have reported increased detection of CMBs with increased MRI field strength. (3) Discuss more extensively the potential that

Comment 2:

Utilization of 1.5T as “gold standard” is problematic when we know that higher magnetic field strength MRI such as 1.5T or 7TZ are better for CMB detection. It may be appropriate to simply compare 0.55T to 1.5 T and not report 1.5T as the “gold standard”.

 For example, Conijn et al showed that CMB detection is better with 7T MRI compared to 1.5T.

doi: 10.3174/ajnr.A2450

Author Response

Please see the attachment (Word-doc).

Round 2

Reviewer 2 Report

The authors have addressed my prior concerns.